# Time-Varying Gene Expression Network Analysis Reveals Conserved Transition States in Hematopoietic Differentiation between Human and Mouse

**DOI:** 10.3390/genes13101890

**Published:** 2022-10-18

**Authors:** Shouguo Gao, Ye Chen, Zhijie Wu, Sachiko Kajigaya, Xujing Wang, Neal S. Young

**Affiliations:** 1Hematopoiesis and Bone Marrow Failure Laboratory, Hematology Branch, National Heart, Lung, and Blood Institute, National Institutes of Health, Bethesda, MD 20892, USA; 2Department of Mathematics and Statistics, Northern Arizona University, Flagstaff, AZ 86011, USA; 3Division of Diabetes, Endocrinology, and Metabolic Diseases (DEM), National Institute of Diabetes and Digestive and Kidney Diseases, National Institutes of Health, Bethesda, MD 20817, USA

**Keywords:** single-cell RNA sequence, time-varying network, transition state during differentiation

## Abstract

(1) Background: analyses of gene networks can elucidate hematopoietic differentiation from single-cell gene expression data, but most algorithms generate only a single, static network. Because gene interactions change over time, it is biologically meaningful to examine time-varying structures and to capture dynamic, even transient states, and cell-cell relationships. (2) Methods: a transcriptomic atlas of hematopoietic stem and progenitor cells was used for network analysis. After pseudo-time ordering with Monocle 2, LOGGLE was used to infer time-varying networks and to explore changes of differentiation gene networks over time. A range of network analysis tools were used to examine properties and genes in the inferred networks. (3) Results: shared characteristics of attributes during the evolution of differentiation gene networks showed a “U” shape of network density over time for all three branches for human and mouse. Differentiation appeared as a continuous process, originating from stem cells, through a brief transition state marked by fewer gene interactions, before stabilizing in a progenitor state. Human and mouse shared hub genes in evolutionary networks. (4) Conclusions: the conservation of network dynamics in the hematopoietic systems of mouse and human was reflected by shared hub genes and network topological changes during differentiation.

## 1. Introduction

Throughout life, hematopoietic stem cells (HSCs) maintain the mammalian blood system [1]. HSCs have three major functional attributes: self-renewal to maintain a stem cell pool; differentiation along distinct lineage pathways; and proliferation. Blood diseases can result from an imbalance of fate choices and biased productions of cell types. A cellular fate is mainly determined by activation of specific transcription factors and their target genes in complex transcriptional regulatory networks [2]. Identification of valid transcriptional regulations in biological processes with experiments remains challenging, and many network reconstruction algorithms have been developed to infer functional relationships between gene pairs. Most approaches have been based on bulk expression profiles for samples which contain highly diverse cell types. Network reconstruction with time-series data has become popular because the data capture a more thorough picture of the system than does non-temporal data [3,4]. Recently, single-cell RNA sequencing (scRNA-seq) has provided a powerful method to discover regulatory relationships in hematopoiesis [5,6,7]. Pseudo-time ordering places individual cells along a virtual time axis and provides a large amount of complex additional information for network analysis. Gene interactions can change over time, and such changes can be inferred from time-varying data. As an illustration, networks of three genes at three time points were not identical (Figure 1a) [8]. It is biologically meaningful to examine the evolution of these patterns over time rather than characterizing a single static graph that represents only the interactions persistent over time [6,7]. Reconstruction and dynamic network analysis with bulk or single-cell data help to understand evolution of biological processes (differentiation, development, and disease onset and progress) at the network level [3,4]. It is of more interest to investigate the critical transition state as well as the genes that control the transition with changes of networks across time. The temporal dynamics of this intergenic interaction during hematopoietic differentiation are yet to be delineated. Gaussian Graphic Models (GGMs) have been successfully applied to time-series data to estimate time-varying graphs, on the assumption that covariance matrices change smoothly over time [9,10,11,12,13]. Thus a series of networks across time facilitate understanding of the evolution of gene interactions. Recently, a Local Group Graphical Lasso Estimation (LOGGLE) [14] model has been proposed to incorporate network gradual topology alteration over time, in order to estimate edge sets of networks at different time points. In this model, neighborhood information is efficiently integrated, and a computational speed is greatly increased by a block-wise fast algorithm and pseudo-likelihood approximation. LOGGLE has been utilized to model direct dynamic interactions between stocks during the global financial crisis [14].

Here, we apply the LOGGLE model to pseudo-time ordered gene expression dataset of three lineage differentiation branches of human and mouse so as to construct hematopoietic time-varying gene interaction networks, since the model fits the biological realm of differentiation. We quantify an evolutionary trend of the gene network with differentiation (ordered by pseudo-time) by examining global attribute indicators, such as network density. We further apply network similarity analysis to confirm three stages of differentiation and to identify hub genes at different stages using centrality analysis. Last, we examine evolutionary changes of gene network topology as they can provide novel insights into hematopoietic differentiation. Hub genes identified in the aggregated networks (combined networks at all times via a union operation) at different evolutionary stages provide interacting candidate genes for further studies. Our model effectively captures the structural transitions in the dynamic networks.

## 2. Materials and Methods

### 2.1. scRNA-seq Data from Hematopoietic Stem and Progenitor Cells (HSPCs) of Human and Mouse

Bone marrow samples were obtained from healthy donors, as described in a previous study [7]. CD3^−^CD14^−^CD19^−^CD34^+^ cells were sorted using a LSRII Fortessa Cytometer. Lineage^−^CD117^+^ cells from bone marrow of C57BL/6 mice were sorted. scRNA-seq cDNA libraries for human and mouse were prepared with the Chromium Single Cell 3‘ platform (10x Genomics, Pleasanton, CA, USA). scRNA-seq libraries were sequenced on the Illumina HiSeq 3000 System. The cellranger pipeline was used to process raw data, to align reads to the genome, and to produce gene–cell expression matrices [15].

### 2.2. Preprocessing of Gene Expression Data

The R software package Seurat was used for downstream analyses (Figure 1b) [15]. Raw reads in each cell were first scaled to 10,000 and log-transformed. Highly variable genes were identified with the FindVariableGenes function for Principal Component Analysis (PCA). Unsupervised clustering of cells was performed with a graph-based clustering approach at resolution 2, based on the top 30 principal components. Cells were visualized based on uniform Manifold Approximation and Projection (UMAP) in Seurat. For each cluster, gene expression was compared to a median expression of the same gene from cells in all other clusters by the FindMarkers function in Seurat and cluster-specific genes were identified with *p* < 0.01 as the cutoff. Genes then were ranked based on their expression fold changes, and top cluster-specific genes were compared with published cell type-specific genes. An HSPC subtype of each cluster was assigned based on statistical significance of overlap between HSPC- and cluster-specific genes (Fisher’s exact test). Expression analysis for mouse followed the same pipeline, using cell lineage-specific genes derived from GSE81682 in GEO as references for cell type assignment.

Differentiation trajectory analyses were conducted with Monocle 2 [16]. Preprocessed Seurat objects were imported into Monocle 2 with the “importCDS” function. The Monocle 2′s “orderCells” function arranged cells along a pseudo-time axis to indicate their positions in a developmental continuum. A reversed-graph embedding algorithm was used to impute differentiation trajectories and infer pseudo-time of cells.

### 2.3. Selection of the Most Relevant Genes

We first filtered genes to reduce noise and data dimensionality. In this work, we considered the following two criteria for feature screening to obtain genes that carry important information (Figure 1b), by merging the gene lists from manual annotation and expression atlas of hematopoiesis [17,18].

(1) Relevance to hematopoiesis. We used an annotated gene list from one report [17]. There were 45 genes in total, including 33 transcription factors important for HSCs and hematopoiesis, and 12 additional genes implicated in HSC biology, after removing three housekeeping genes [17]. Homologous genes of human were obtained accordingly (http://www.informatics.jax.org/, accessed on 1 January 2022).

(2) We also retrieved the lineage-specific genes (progenitors only) from Haemopedia, an atlas of murine gene-expression data of 54 hematopoietic cell types [18]. We calculated variance for each gene and selected top genes to construct time-varying network graphs. Genes with smaller variance (not significantly expressed in any lineage with a cutoff of standard deviation > 2.5 [18]) did not contain much information across the lineages and thus were not considered. A purpose of this was to explore the evolution of networks constructed with genes showing high variations in different differentiation stages. For human, genes with maximal expression in the orthologous lineage were obtained for network reconstruction [18]. The gene list is given in Appendix A.

### 2.4. Estimating Time-Varying Graphs with the LOGGLE Model

We aimed to characterize an evolutionary pattern of inter-gene interactions over time during the differentiation and to identify hub genes involved in network evolution. Accordingly, we first used the LOGGLE model developed by Yang and Peng [14] (https://github.com/jlyang1990/LOGGLE, accessed on 31 November 2021) to build and understand differentiation time-varying network graphs (Figure 1c). For the gene expression data, due to high dropout rates of single-cell data, for each branch we binned ordered cells into 25 bins and used the averaged gene expression of cells in the same bin to represent expression at certain time points. The model supposes that graph topology changes smoothly over time, and it uses a local group lasso type penalty to represent information from adjacent time points to ensure a smooth change in the graph structure. To make the work self-contained, we here describe how the LOGGLE model constructs a time-varying network graph. More technical details are described in its original paper and in the LOGGLE package [14].

### 2.5. Local Group Graphical Lasso Estimate

For a p-dimensional time-series random vector $X\left(t\right)=\left(X^{1}\left(t\right),\;X^{2}\left(t\right),\ldots ,\;X^{p}\left(t\right)\right)$ at time $t\in \left[0,1\right]$, which follows a multivariate Gaussian distribution $N_{p}(\mu \left(t\right)$, $\sum \left(t\right))$. $\{x_{k}\}\;\left(k\in \left\{1,\ldots ,N\right\}\right)$ indicates the observation at time $t_{k}\left(0\leq t_{1}\leq \cdots \leq t_{N-1}\leq t_{N}\leq 1\right)$, in which N  represents the number of time points. In our analysis, p is the number of genes.

LOGGLE aims to construct the graph edge set by estimating the precision matrix $\Omega \left(t\right)=\Sigma^{-1}\left(t\right)$. The model assumes the smoothness of the graphical profile, that is, the edge set of the estimated network changes gradually over time. This is achieved by penalizing with difference between the adjacent networks in the function. The output of the model is d precision matrix $\overset{´}{\Omega (}t_{k})$ at the kth time point with the local group lasso penalty, through combining the locally weighted negative log-likelihood function [14]:(1)$$L\left(\Omega_{k}\right)∶=\frac{1}{\sqrt{\left|N_{k,d}\right|}}\sum_{i\in N_{k,d}}\left[tr\left(\Omega \left(t_{i}\right)\hat{\Sigma }\left(t_{i}\right)\right)-\log\left|\Omega \left(t_{i}\right)\right|\right]+\lambda \sum_{\mu \neq \nu }\sqrt{\sum_{i\in N_{k,d}}\Omega_{\mu \nu }{\left(t_{i}\right)}^{2}}\;$$
where $N_{k,d}=\left\{i\in 1:\left|t_{i}-t_{k}\right|\leq d\right\}$ works as the penalty at the center $t_{k}\;$ and neighborhood width d; $\left|N_{k,d}\right|$ is for normalization of $N_{k,d}$.

### 2.6. Model Fitting and Parameter Adjustment

In the algorithm of LOGGLE, three parameters need to be determined: the bandwidth of kernel estimation h; the neighborhood width d, which controls the smoothness of the graph over time; and the sparsity parameter λ, which determines the degree of graph sparsity. The model uses the alternating directions method of multipliers (ADMM) algorithm [11] to obtain the optimized results for the objective function in Equation (1). Through cross-validation (*CV*), parameters at each time point were calculated (Figure 1c). Specifically, data were divided into training and validation sets, and a validation score on a *j*th validation set was calculated by:(2)$$CV_{j}\left(t_{k};\;\lambda_{k};\;d_{k};h\right)=tr\left({\hat{\Omega }}_{-j}^{rf}\left(t_{k};\;\lambda_{k};\;d_{k};h\right)\;{\hat{\sum }}_{\left(j\right)}\left(t_{k}\right)\right)-log\left|{\hat{\Omega }}_{-j}^{rf}\left(t_{k};\;\lambda_{k};\;d_{k};h\right)\right|$$
where $CV\left(t_{k};\;\lambda_{k};\;d_{k};h\right)=\overset{K}{\underset{j=1}{\sum }}CV_{j}\left(t_{k};\;\lambda_{k};\;d_{k};h\right)$) is the *K*-fold cross-validation score at time $t_{k}$. The optimal combinations of three parameters $\left(h,\;d_{k},\lambda_{k}\right)\;$ are the values that yield the smallest CV score. To reduce a false positive rate, the “majority vote” procedure cv.vote was calculated. A flow chart of LOGGLE algorithm is shown in Figure 1c.

### 2.7. Global Network Properties

Comparing network properties can provide good insights for interacting relationships of genes within biological networks in a timely manner. For a network G (V,E), in which V and E are vertices and edges, several common network properties, including the number of edges, network diameter, and network density, were examined to explain a trend of network topology changes. The network diameter represents the shortest distance between the two most distant genes in the network, calculate as, $D=\underset{i,j}{\max}\delta_{min}\left(i,j\right),$ where $\delta_{min}\left(i,j\right)$ represents the shortest path between gene i and j. A higher diameter indicates that compactness between nodes in the network is low. The network density describes a portion of potential connections in the network that are actual connections, and is defined as $d\left(G\right)=\frac{2\left|E\right|}{\left|V\right|\left(\left|V\right|-1\right)}$.

### 2.8. Network Similarity Analysis

After obtaining a series of networks at different time points, the similarity between networks will be examined and similar networks will be merged for biological interpretation. CompNet neighbor similarity index (CNSI) is used to measure the similarity between two compared networks [19]. Given two networks A and B, the similarity of each pair of genes is calculated with the degree of overlap between the first neighbors of the nodes in the two networks $\;\mathrm{with}\;CNSI_{i}=\frac{f_{n_{i}}^{A}\cap f_{n_{i}}^{B}}{f_{n_{i}}^{A}\cup f_{n_{i}}^{B}}$, where $n_{i}$ is the i-th gene of the two networks, and $f_{n_{i}}^{A}\;\mathrm{and}\;f_{n_{i}}^{B}\;$ refer to the first neighbors of the *i*-th gene in the two compared networks. The similarity between the two networks is represented by the sum of $CNSI_{i}$ of all genes.

### 2.9. Centrality Analysis

In a gene network, not all genes equally influence a network. Gene networks usually follow a scale-free distribution, in which the majority of the genes have one or two connectivities, and only a few genes have large numbers of connectivities. Many centrality measures are proposed to indicate a gene’s importance in the network context, such as connectivity/degree (the number of first neighboring genes), betweenness (frequency of genes is passed by the shortest paths of pairs of all other genes), clustering coefficient (probability of connections among gene’s direct neighbors), and PageRank (popularity of a gene based solely on the interactions). These measures were calculated with igraph (https://igraph.org/r/, accessed on 22 January 2022). We also borrowed the concept of the h-index, as widely used in the publication citation Networks [20]. The h-index of a scientist is defined to be x, if one has published at least x papers with x or more citations each, and is designed to capture both productivity and impact of published work. A recent study found that in a number of manmade networks, the h-index performs well at capturing the spreading influence of genes when compared with gene degrees in a network. The h-index was calculated with the influential package (https://github.com/asalavaty/influential, accessed on 25 November 2021).

### 2.10. Characterization of a Global Network with Concepts of Entropy and Energy

The centrality features of all genes can be used to characterize the global network. As a measure of uncertainty, the entropy captures the amount of information lacked in a system, as a measure of uncertainty [21]. The more deterministic the network, the smaller the uncertainty in the configuration of the graph is, and the smaller the entropy of the graph. Overall certainty can be also assessed with the Shannon entropy of this network, given by:$$H\left(Q\right)=-1*\overset{n}{\underset{i=1}{\sum }}q\left(k\right)*logq\left(k\right)$$
where $q\left(k\right)=value\left(k\right)/sum\left(value\left(k\right)\right)$, and the value is degree, betweenness, or other measures of a gene k. $H\left(q\right)$ provides a measure of the network’s heterogeneity in degree or betweenness of genes.

The concept of graph energy has been subjected to research in the domains of chemistry, physics, and complex networks [22]. Graph energies reflect the compositions of subgraphs in the network. Graph energies are matrix energies of various graph representations and can be defined for any symmetric graph matrix. Graph energy can be defined over the adjacency matrix, Randic, and Laplacian energies are defined over the Randic and Laplacian matrices [22,23].

Graph energy is defined on the basis of the adjacency matrix $M_{A}$ of the network, let
$$M_{A}\left[i,j\right]=\left\{\begin{array}{c} 1\;\;if\left\{v_{i},v_{j}\right\}\in E\\ 0\;\;\;\;otherwise \end{array}\right.$$
be the adjacency matrix of *G* with nodes (*V*) and edges (*E*). The energy is defined as:$$E_{G}\left(G\right)=\overset{n}{\underset{i=1}{\sum }}\left|\mu_{i}\right|$$
where $\mu_{1},\ldots ,\mu_{n}$ are the eigenvalues of the adjacency matrix $M_{A}$.

The Randic adjacency matrix $M_{R}$ of the network is defined as:$$M_{R}\left[i,j\right]=\left\{\begin{array}{c} 0\;\;\;\;if\;i=j\\ \frac{1}{\sqrt{C_{D}\left(v_{i}\right)*C_{D}\left(v_{j}\right)}}\;if\left\{v_{i},v_{j}\right\}\in E\\ 0\;\;\;\;if\;\left\{v_{i},v_{j}\right\}\notin E \end{array}\right.$$
where $\;C_{D}\left(v_{i}\right)$ and $\;C_{D}\left(v_{j}\right)$ are the degrees of $v_{i}\;$ and $v_{j}$. The Randic energy is defined as:$$E_{R}\left(G\right)=\overset{n}{\underset{i=1}{\sum }}\left|\rho_{i}\right|$$
where $\rho_{1},\ldots ,\rho_{n}$ are the eigenvalues of the adjacency matrix $M_{A}$.

Laplacian energy is defined on the basis of the Laplacian matrix $M_{L}$ of the network, Let,
$$M_{L}\left[i,j\right]=\left\{\begin{array}{c} d_{i}\;\;\;\;\;\;\;\;\;\;\;\;\;\;\;\;\;\;\;\;\;\;\;\;\;\;\;\;\;\;\;\;\;\;if\;i=j\\ -1\;\;\;\;\;\;\;\;\;\;\;\;if\;i\neq j\;\cap \left\{v_{i},v_{j}\right\}\in E\\ 0\;\;\;\;\;\;\;\;\;\;\;\;\;\;\;\;\;\;\;\;\;\;\;\;\;\;\;\;otherwise \end{array}\right.$$

The Laplacian energy of *G* is defined as:$$E_{L}\left(G\right)=\overset{n}{\underset{i=1}{\sum }}\left|\lambda_{i}-\frac{2m}{n}\right|$$
where $\lambda_{1},\ldots ,\lambda_{n}$ are the eigenvalues of the adjacency matrix $M_{L}$. *m* and *n* are the numbers of edges and nodes.

## 3. Results

### 3.1. scRNA-seq Identified a Comprehensive and Conserved List of HSPC Types

Bone marrow samples from four human donors were collected to characterize the early stages of hematopoiesis. Lineage^−^CD34^+^ cells were sorted to enrich for HSPCs. A human dataset contained 15,245 single CD34^+^ stem/progenitor cells after filtering out cells with small numbers of detected genes, as visualized in UMAP, displayed clear clusters, suggesting distinct cell types at molecular levels (Figure 2a). Hematopoietic cell identity was assigned to each cell cluster by examining cluster-specific genes with a reported lineage signature gene list [1,6]. CD34^+^ cells were grouped into 15 clusters and then computationally assigned to the following cell populations: hematopoietic stem cells and multipotent progenitors (HSCs), granulocyte–monocyte progenitors (GMPs), B megakaryocyte–erythroid progenitors (MEPs), lymphocyte progenitors (ProBs), and early T lineage progenitors (ETPs). With the same computational strategy, 17,560 lineage^−^CD117^+^ cells from B6 mice were also clustered, unsupervised, based on transcriptome similarity using UMAP. Hematopoietic cell identity was assigned to each cluster of cells by comparing cluster-specific genes with published lineage signature genes. Cells were grouped into 36 clusters and assigned into long-term hematopoietic stem cells (LTHSCs), multipotent progenitors (MPPs), lymphoid multipotent progenitors (LMPPs), common myeloid progenitors (CMPs), MEPs, and GMPs (Figure 2a). LMPP in mouse corresponds to ProB and ETP populations in human, and they were used to be compared in the appendix of this paper.

### 3.2. Differentiation Trajectories in Human and Mouse Hematopoiesis

Clustering assumes biologically distinct groups, such as discrete cell types or states, and pseudo-temporal ordering assumes that data lie on a connected manifold. A Waddington landscape helps to illustrate progressive restriction of cell differentiation. Cells traverse a landscape of valleys separated by ridges. The ridges prevent spontaneous conversion of cell types, and thus a cell’s fate is restricted once it descends into a specific valley (Figure 2b). Pseudo-temporal ordering helps to map the pathways in such a landscape and the underlying regulatory programs along the paths, revealing how cell types are stabilized with differentiation. We used Monocle 2 to arrange each cell by pseudo-temporal ordering, in order to analyzing the transition from stem cells to lineage-restricted progenitors. After applying Monocle 2 to profiled human and mouse cells, an intuitive graphical representation of early stages of HSPC differentiation emerged. In human, lineages clearly separated among lineage^−^CD34^+^CD38^+^ progenitors (Figure 2a).

We defined HSC in human and LTHSC in mouse as roots, so that they were located at starting points of the hierarchies. In both human and mouse, three branches arose from HSC and LTHSC. When the three-dimensional projection of Monocle 2 was colored with assigned cell types, conservation of hematopoietic differentiation between human and mouse was evident. Both human and mouse cells were distributed along pseudo-temporally ordered paths from HSCs/LTHSCs to three branches: erythroid/megakaryocytic, myeloid, and lymphoid (Figure 2c). Since the adjacency of cell types on plotting reflects differentiation pathways at molecular levels, we concluded differentiation trajectories of human and mouse were highly similar, indicating the species’ conservation.

### 3.3. Evolution of Time-Varying Network Graphs during Hematopoietic Differentiation

When the parameter d is set to 0, the network does not change over time. In contrast, when d ≠ 0, the differentiation time-varying graph constructed by the LOGGLE and kernel models captures evolving patterns of gene interactions over time. Thus, both the LOGGLE and kernel models can describe the evolutionary mode of differentiation well, but the LOGGLE model has a better CV score and is better supported by data. Finally, we chose the well-balanced LOGGLE model for further analysis. Six models for the three branches of human and three branches of mouse were obtained. Appendix A shows a result of the parameter selection of the LOGGLE model, obtained by cross validation, at each differentiation stage.

Based on the above model comparison results, we selected the best performing LOGGLE model results to further analyze the evolution of the hematopoietic differentiation gene expression networks over time. A time-varying graph of the gene interaction networks fitted by this model, at 25 pseudo-time points, were obtained for all three branches of human and mouse. For a list of network edges corresponding to all differential stages, refer to Appendix A. Note that some interactions among genes only appeared in certain time points. For example, in mouse interactions among *Slc4a1*, *Fn3k*, *Alas*, and *Hbq1i* only appeared at time 24. An interaction between *BPTF* and *HBX1* in the human network disappeared in late stages (Appendix A).

To illustrate changes in network topology in more detail, we calculated several global network properties. It was observed that, from the HSC stage (stable state) to the transition stage with differentiation, complexity of a gene interaction network (calculated with network density) began to decrease, bottoming in the transition stage, then increased again before becoming progenitor cells (another stable state) (Figure 3a). It is interesting that in three branches of both human and mouse, U shapes of network density were observed. This may suggest that transition stages are subject to weaker regulatory constraints.

This phenomenon was consistent with the themes of human and mouse differentiation, proving advantages of single-cell transcriptome profiling, which allowed inspection of cell states and cell-state transitions at fine resolution, and the identification of transition cells [24]. Transition cells were characterized by their transient dynamics during a cell-fate switch, or their mixed identities from multiple cell states, different from the well-defined stable cell states that usually express marker genes with distinct biological functions (Figure 3b).

Dynamic modeling provided a method to characterize multi-phase cell-fate transitions (Figure 3b). There are at least three possible perspectives to describe cell-fate transitions, as either entirely discrete or continuous process, or as a multi-phase switch process between two stable states mediated by the transition cells. The first two perspectives corresponded to clustering or pseudo-time ordering frequently adopted in single-cell analysis (Figure 3b). In the multi-phase model, cells undergoing transitions were analogized to particles, in which the transient states corresponded to saddle points and the stable cell states corresponded to commonly observed cell populations (Figure 3c).

The number of cells in each state depends on the stability of the state, which is determined by the energy states of the cells within the state. More cells would be found in stable states and less cells in transition states, due to different levels of stability (Figure 3c). To confirm the properties of transition cells, we examined expression changes of signature genes over pseudo-time. Expression of both *Gata1* in mouse and *GATA1* in human graphs were sigmoid shapes, implying existence of transition cells (Figure 3d,e). Cells in stable states 1 and 2 formed two types of populations, which could be separated computationally or experimentally. Transition cells made pseudo-time ordering possible because they provided a bridge between two stable states.

The gene expression change (Figure 1e) was not synchronized with the transition state identified by network density (Figure 1a). This is because the correlation or the gene-gene interaction may not happen at the time of the highest gene expression, or with synchronized modes, as the description for the model of FeedForward Loop [25]. Regulation may happen at certain time, leading to the increase in gene expression, and there is a time delay. Additionally, gene regulation can only initiate with sufficient gene expression and accumulation of protein regulators.

We further analyzed the similarity between the networks at different stages of differentiation using the CNSI indicator. Corresponding hierarchical clustering (Figure 4a) aggregated 25 time points of networks from HSC to ProB into three stages. A CNSI chart also revealed that the similarity among the adjacent HSC stages was very high (with high CNSI). Similarity among progenitor stages was also high. In contrast, similarity between the transition stage and HSC/progenitor stage was low, and a dendrogram of clustering placed the network in three separate clusters. Network similarity analysis showed that the differentiation was divided into three stages: a prime differentiation stage, a differentiation transition stage, and a progenitor stabilized stage. Although the differentiation was continuous, clusters were formed due to the existence of the stable states of HSC and progenitors.

### 3.4. Estimated Time-Varying Networks Were Robust to the Choice of Pseudo-Time Construction Tool

More than 70 pseudo-time construction tools had been already published until 2019 [26], and they provide different algorithms to estimate pseudo-time with single-cell data. To examine the robustness of our conclusions, we used SlingShot [27], which is able to handle complicated lineages, to recompute pseudo-time for time varying network analysis. HSC in human and LTHSC in mouse were used as starting clusters for SlingShot analysis. We selected the value with the largest weight to assign pseudo-time, according to the suggestion from the SlingShot website.

First, the correlation of estimated pseudo-times between Monocle 2 and SlingShot was relatively high (r = 0.995 to MEP, r = 0.990 to GMP, and r = 0.994 to LMPP; Spearman correlation) in mouse, and also high in human (r = 0.495 to MEP, r = 0.740 to GMP, and r = 0.640 to ProB). The relatively lower correlations in human may be due to the heterogeneity of different individuals (four healthy donors in this study) and complicated lineage specifications in human [1]. Then, pseudo-time ordered cells by SlingShot were analyzed with LOGGLE, and the results were compared with those from Monocle 2-based results. Overlapping interactions were 20–30 times higher than by random chance in mouse, and 8–15 times higher than random chance (*p* values < 1 × 10^−5^ in all time points, *t*-test) in human (Figure 3f). Considering the current challenge in network reconstruction with single-cell data, the consistency was quite high [2]. The existence of transition states was also observed with a SlingShot approach, and we found weaker strains of gene regulation through analyzing changes of global characteristics (density of network) in transition states.

### 3.5. Structural Measures of Transcription Regulation Networks of Genes Involved in Hematopoiesis

The time-varying networks in all three branches showed good scale-free behaviors, for both human and mouse (Appendix A). Frequency of connectivity had a negative logarithmic correlation with the connectivity. In all networks, most genes were connected to only a few other genes, showing a hallmark of scale-free networks [28]. The test statistic of a Kolmogorov-Smirnov test (with the power.law.fit function in the igraph package) that compares the fitted distribution with the degree distribution is shown in Appendix A.

Compared with random networks, time-varying networks had lower entropy (Figure 5a,b). A network with a uniform topology would have maximum degree entropy. One implication was that significant heterogeneity existed among genes in the network evolving through non-random processes during hematopoiesis, and high-degree or high-betweenness genes.

Network energy is an invariant that encodes the network structure. It is defined as a sum of absolute eigenvalues of a matrix, and so it is closely related to the network structure [29]. Moreover, the dominating eigenvalues of adjacency matrix, Randic matrix, and Laplacian matrix are proved to be related to network invariants, such as the largest node degree, connectivity, the number of short cycles, and paths [29]. Time-varying networks had smaller graph energies (Figure 4a,b). The graph energy is calculated by summing the traces of the even powers of the adjacency matrix [29]. Using this new representation, new bounds for the energy are sums of contributions of subgraphs. Consequently, based with this structural interpretation, graph energy can be used in the general context of structural graph theory or even to study gene networks [29]. Interpretation of the graph energy can now be used to assess some real-world graphs, in which the specific contribution of subgraphs can be obtained more precise bounds of the energy as the sum of fragments’ contributions, such as a subgraph containing a square with a pendant vertex and a subgraph containing two triangles with a shared vertex.

### 3.6. Hub Genes Accompanying the Differentiation in Hematopoiesis

Generally speaking, topological features of genes in a network associate to their biological importance [28]. Genes with high connectivity are termed “hub genes” and are usually functionally important. “Betweenness” measures the number of the shortest paths transiting through the genes, and the highest betweenness genes control most of the information flow in the network, and thus representing its critical nodes of the network. Betweenness is a better indicator of essentiality than is gene connectivity, but they are usually highly correlated. Network connectivity and betweenness of our gene lists are shown in Appendix A, and individual examples among genes with top degree and betweenness are provided below.

*MEIS1* expression is correlated with cell self-renewal in normal hematopoiesis, and its expression level is highest in HSCs and declining with differentiation. In mouse, *Meis1* is required to maintain functional LTHSCs [30]. *Gata1* is a hub gene in time-varying network from HSC to MEP with high betweenness. Gene targeting studies of *Gata1* have confirmed its importance in primitive and definitive erythroid cells and megakaryocytes. For examples, in chimeric mice, *Gata1*-null erythroid cells are not able to mature beyond the proerythroblast stage, and a lack of *Gata1* in megakaryocytes leads to increased proliferation and deficient maturation of megakaryocytic progenitors [31]. *Gata2* is a hub gene with high betweenness in the LMPP subnetwork. (*Gata2* is critical in also stem cells, and transcriptional gene expression cannot distinguish LMPP from HSC [1,7]).

### 3.7. Conservation of Time-Varying Networks between Human and Mouse

Animal models are widely used in biological research on the predicate that fundamental biochemical processes are conserved across species, as between human and mouse [9]. Evolutionary cross-species comparisons can provide a framework to refine human biological research. scRNA-seq has been extensively applied to study hematopoiesis of human and mouse, but cross-species comparison of the hematopoietic system is not firmly established at network-level comparisons. We collapsed the networks of different times to obtain an aggregated network and calculated an average of all central indices. A range of gene connectivity values in human were generally comparable to those in mouse, suggesting broad structural similarity in gene regulation (Figure 4b). There was high correlation in genes’ connectivity values between human and mouse networks (Figure 4b). PageRank and betweenness centralities confirmed high correlation of human and mouse (Figure 4b). These results suggested there was species conservation and genes with strongly conserved connectivity were generally to be functionally evolutionary stable, and played important roles in hematopoiesis. In addition to conserved network centrality between human and mouse, the time-varying networks of two species share other same global characters. Both showed scale-free behaviors that the degree distribution was a power-law. Calculation of the shortest path length (L) and clustering coefficient (CC), and comparison with the randomized networks with the same number of nodes and edges showed that networks owned small-world properties (L  ≈  L random, CC  ≫  CCrandom) [9].

### 3.8. Conserved Networks between Human and Mouse

Networks were converted into hypergraphs with the Dual Hypergraph Transformation (DHT) approach, which transforms the edges of a graph into the nodes of a hypergraph (Figure 6a). The total number of gene-gene interactions in the 25 original time-varying networks were assigned, as attributes to nodes of a hypergraph (the number of edges between associated two genes) [32]. There were a high number of overlapping interacting gene pairs appearing both in human and mouse networks, with odds ratios of 1.7, 2.2, and 3.1; and *p* values of 3 × 10^−4^, 8 × 10^−6^, and 3 × 10^−3^, comparing to the numbers occurred by chance in HSC to GMP, HSC to MEP, and HSC to LMPP. For the total number of interactions (the number of appearances in all times) of shared interacting gene pairs, correlations were high between human and mouse in three trajectories (HSC to MEP, r = 0.42, *p* = 0.019; HSC to GMP, r = 0.24, *p* = 0.06; and HSC to LMPP, r = 0.26, *p* = 0.05). These showed species’ conservation of gene regulation during hematopoiesis.

Considering conservation between human and mouse networks, we applied an R-package, bioNet, to three aggregated hypergraphs for the analysis with a heuristic approach to identify sub-hypergraphs which had higher numbers of edges in 50 time-varying networks in human and mouse [33]. Three sub-hypergraphs were converted back to gene modules, and these modules played critical roles in the differentiation (Figure 6b). In the module from HSC to MEP, *GATA1* and *GATA2* are both hub genes, as expected. *BLNK*, a hub gene for the module from HSC to LMPP, encodes a cytoplasmic linker or adaptor protein that plays a critical role in B cell development [34]. All genes and their related degrees in the identified core subnetworks are shown in Appendix A. In all three networks, most genes were connected to only a few other genes, which is a hallmark of scale-free networks. We examined whether the core networks were small-worlds through generating randomized networks with the same number of nodes and edges, and compared the mean shortest path length (L) and the clustering coefficient [35]. The conserved networks of HSC to GMP, HSC to MEP, and HSC to LMPP had the shortest path lengths of 1.98, 1.76, and 2.61, respectively. The shortest path lengths for Erdos random networks were 1.86 ± 0.004, 1.69 ± 0.001, and 2.32 ± 0.03, respectively. Their clustering coefficients were 0.379, 0.510, and 0.366, respectively, and the same-sized randomized networks had clustering coefficients 0.187 ± 0.007, 0.309 ± 0.006, and 0.134 ± 0.027, respectively. Thus, all three networks had the properties of a small-world with some highly connected subnetworks (Appendix A) [35].

## 4. Discussion

Hematopoiesis is a stepwise process, originating from HSCs and associated functionally with activation of lineage-specific transcription factors for progenitor cells. Transition cells are considered critical in many important biological processes, such as in organ development. We performed time-varying network reconstruction and analysis on pseudo-temporally ordered gene expression data of cells during hematopoietic differentiation. Conservation between human and mouse should help in interpreting disease models for research. Due to the complexity of the LOGGLE algorithm, we could only include about 100 genes, and more efficient algorithms are needed for the analysis of more genes. Another limitation is that the estimated pseudo-time does not accurately represent the biological time, and thus we cannot precisely determine the biologically interesting time points for network reconstruction. Due to the characteristics of single-cell data, a high noise level and a dropout rate affect the performance of LOGGLE to estimate networks, and imputation algorithms to perform denoising and drop out imputation in scRNA-seq may be a good direction to improve the results. The lower consistency between the results with Monocle 2 and SlingShot in human indicates the complicated branching structures may make the pseudo-time ordering difficult, and then affect the network estimations.

## 5. Conclusions

The evolutionary trajectory of the time-varying networks, built with the LOGGLE model, characterizes the changes in transcription programs at the gene interaction levels, instead of at the individual gene expression levels. The differentiation trajectory could be divided into three stages, through similarity analysis of neighboring networks along pseudo-time, and hub genes at different differentiation stages were identified. The evolution of time-varying graphs revealed the differentiation patterns of human and mouse hematopoiesis with three states. Existence of transition states may be able to explain why different phenotype cells can form clusters in UMAP and the cells can be sorted with FACS technology (Figure 3c). Analogous to chemical reactions, the transition can be conceptualized as sharing some features of reactants and products (two stable states), and cannot be isolated due to instability. Identification of the ‘catalysts’ for differentiation would be useful to understand production regulation of blood cells from stem cells. Phase transitions are crucial for the survival and reproduction of the cells, and failure to implement phase transitions will result in dysfunction within cells and networks. It would be desirable to study different models for the network evolution to investigate the regulation change over time. The lower edge density successfully showed weaker regulation in this study, and more quantities rephrased in terms of a nonlinear system on network is expected to investigate the critical transition from healthy states to disease states [36,37]. The investigation of energy in gene networks is still in its infancy. The energy reflects the connections within the network of a vertex define the way in which any abstract resource (information, influence, or importance) circulates in its neighboring vertexes. We report the results here in the hope of stimulating further investigation of network energy [38,39,40,41,42].

There was conservation of the overall hematopoietic process between mouse and human. Both showed three branches of differentiation pathways. Their networks shared hub genes and global topological characteristics. Six dynamic networks all had transition stages with loose network constrains. The conservation of networks and important genes inside helps to understand hematopoiesis and to develop treatment of blood diseases.

## Figures and Tables

**Figure 1 genes-13-01890-f001:**
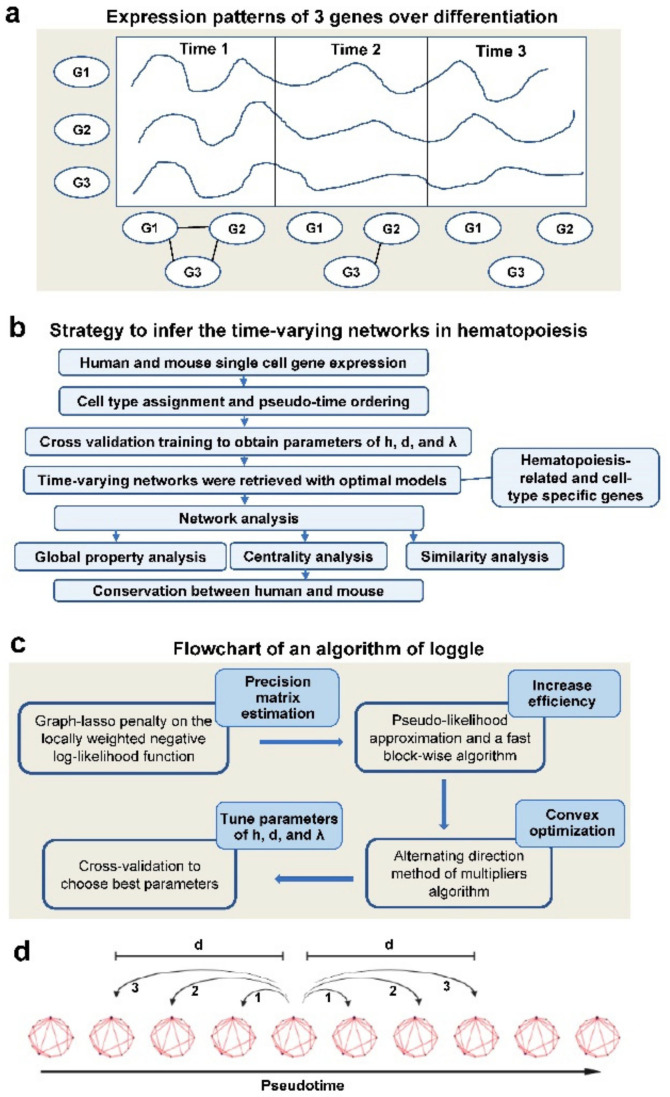
(**a**) Hypothetical model of three genes’ expression patterns with differentiation, from which we inferred a time-varying network and a network change over time. Correlations were inferred from expression at time points and neighboring time points. Three genes were corelated at time point 1. Only G2 and G3 were correlated at time 2, and there was no correlation at time 3. (**b**) Strategy to infer the time varying network in hematopoiesis. (**c**) A flowchart of an algorithm of the LOGGLE, with improvement with the ADMM approach. (**d**) Scheme of neighboring width d for the lasso-type penalty function, pseudo-time is the inferred pseudo-time of the cell at the center of each bin.

**Figure 2 genes-13-01890-f002:**
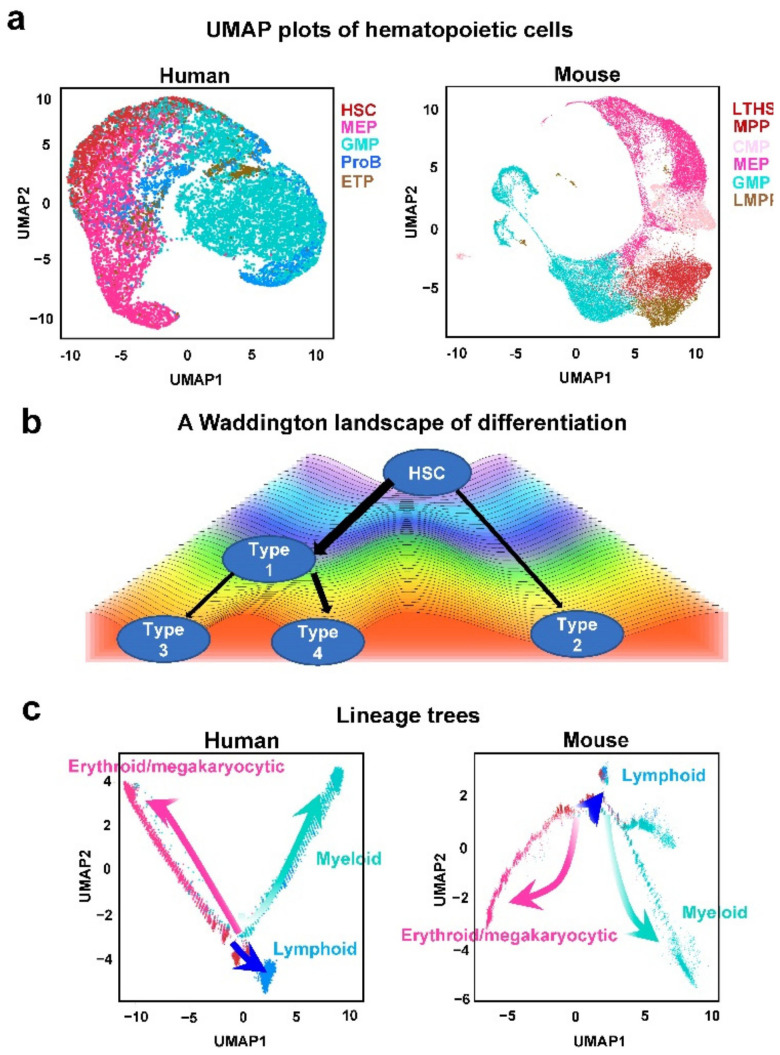
(**a**) UMAP plots of human and mouse hematopoietic cells, colored by cell types. (**b**) A Waddington landscape of differentiation (generated with the program from https://github.com/zzwch/waddingtonplot, accessed on 21 January 2022). A cell fate was restricted at a ridge, and determined once a cell fell into a valley. (**c**) Human and mouse lineage trees inferred by Monocle 2 using reverse graph embedding at two dimensions.

**Figure 3 genes-13-01890-f003:**
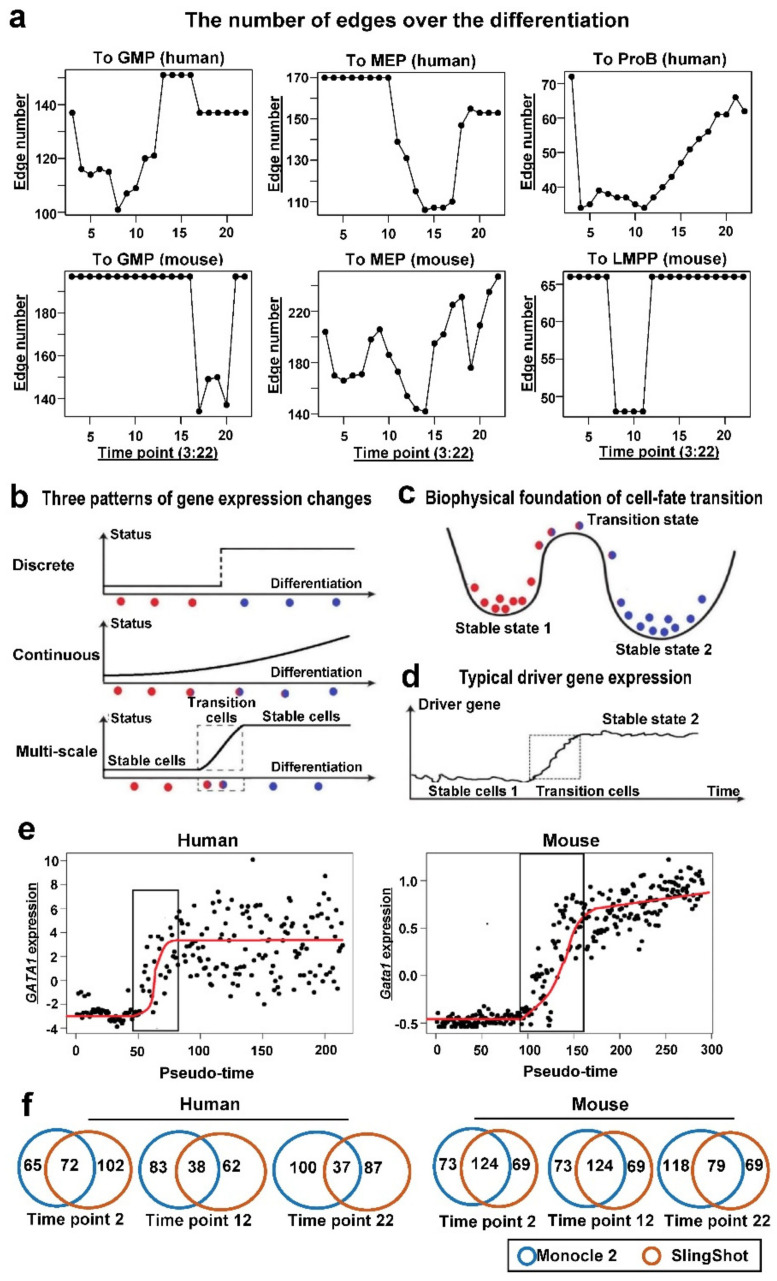
(**a**) The number of edges over the differentiation of three branches. All showed U shapes. “To ProB” in human corresponds to “To LMPP” in mouse, both are for lymphoid direction. (**b**) Three patterns of gene expression changes. (**c**) Biophysical foundation of cell-fate transitions. Stable states corresponded to basins while the transition states were at the saddle points. A transition state (with blue and red colors) owned both characteristics of two stable states, and drifted towards to the second one over time. (**d**) Expression of typical driver genes fluctuated within the stable cells. (**e**) Sigmoid shapes of *GATA1* (human) and *Gata1* (mouse) expression. (**f**) Venn diagrams of overlaps of interactions between Monocle 2- and SlingShot-based approaches.

**Figure 4 genes-13-01890-f004:**
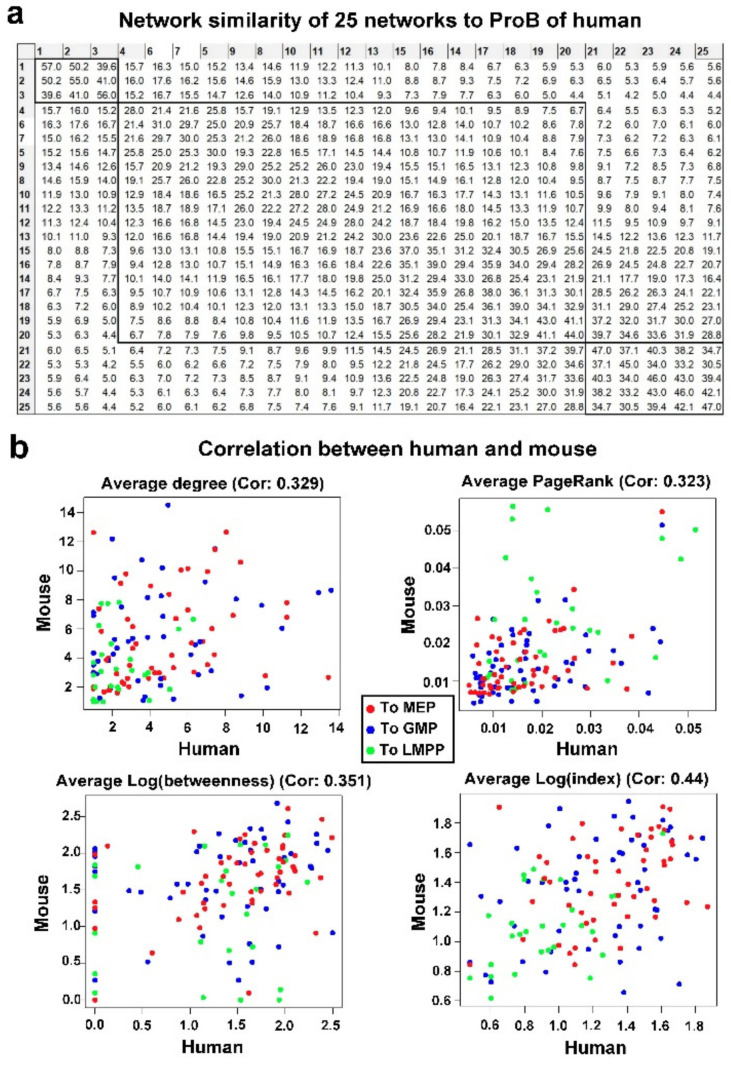
(**a**) Network similarity of 25 networks of human. Values in grids are a sum of CNSI of all genes between two networks. The time points were ordered by hierarchical clustering. (**b**) Correlation of gene connectivity, PageRank, betweenness, and h-index between mouse and human. The Pearson correlations are significant with *p* values of degree: 0.00011, PageRank: 0.00015, Betweenness: 3.355 × 10^−5^, h-index: 1.505 × 10^−7^.

**Figure 5 genes-13-01890-f005:**
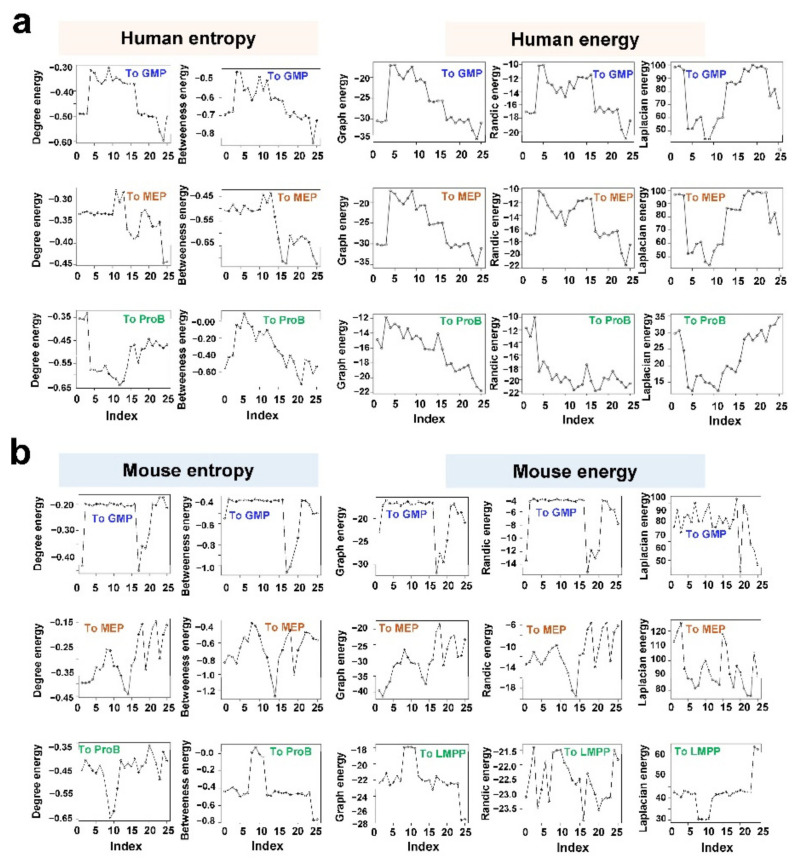
Lower entropy and energy than random networks. Network Entropy and energy of time-varying networks of human (**a**) or mouse (**b**). Random networks with the same nodes and edges were generated, and their entropy and energy were used as references. The *Y* axis shows normalized values against those of random networks.

**Figure 6 genes-13-01890-f006:**
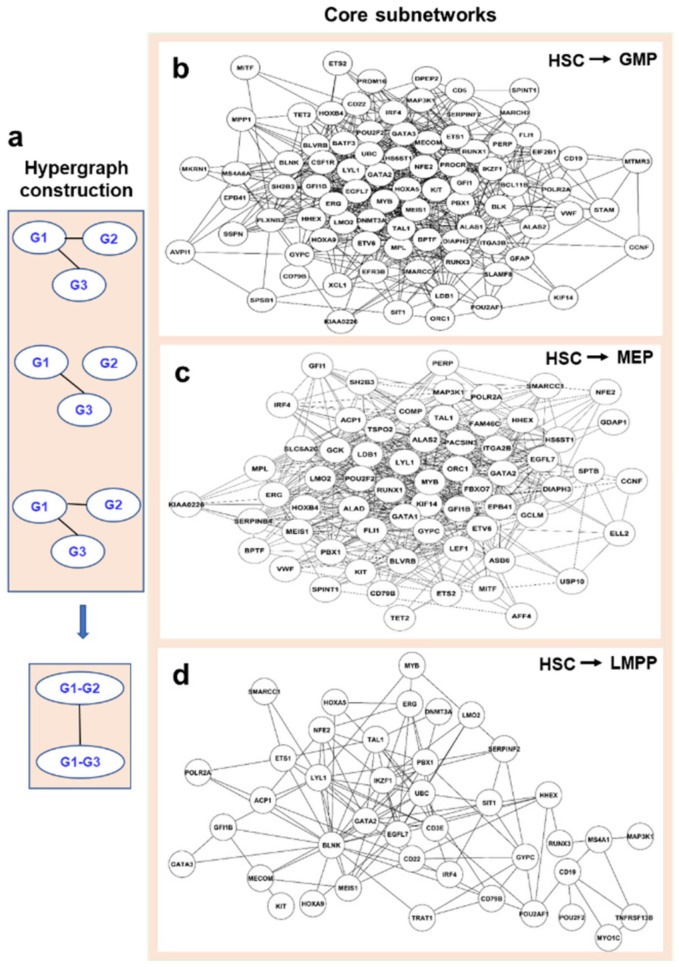
(**a**) A schematic diagram of hypergraph construction. Two nodes were corresponding to two original edges of G1–G3 and G1–G2. G2–G3 did not exist because G2 and G3 did not interact in original networks. The edge number attributes of G1–G3 and G1–G2 are 3 and 2, which will be used as input of BioNet. Core subnetworks expressed from HSC to MEP (**b**), to GMP (**c**), and to LMPP (**d**) progenitors identified by the edge-based scoring approach.

## Data Availability

All data analyzed in this manuscript are already publicly available from the following GEO (www.ncbi.nlm.nih.gov/geo/, accessed on 22 January 2022) accession numbers: (GSE135194 and GSE142235).

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
