# Peer review of "Time-Varying Gene Expression Network Analysis Reveals Conserved Transition States in Hematopoietic Differentiation between Human and Mouse"

_genes, 2022, doi:10.3390/genes13101890_

Round 1
Reviewer 1 Report
This manuscript by Dr Gao et al analyzed the gene-gene coexpression-based network dynamics along hematopoietic differentiation. The authors studied single-cell gene expression profiles of both human and mouse data systematically. The authors also compared the network properties between the human data and the mouse data. The delivery of this study is that the network dynamics are conserved between human and mouse hematopoietic differentiation, in terms of hub genes and some topological changes in the network.
Network dynamics along differentiation is a significant question in biology. However, the novelty of this manuscript is unclear, and several statements need to be more rigorous. Regarding the computational method, I wonder if the authors could discuss the robustness and give more details for their proposed pipeline. The application of structural measurements of transcriptional regulation networks is interesting and maybe novel. However, the interpretation of the results could be clearer. Regarding the biology discovery, if the statement about ‘differentiation trajectories of human and mouse are similar’ is based on Figure 2, then the three major lineages in hematopoietic differentiation (Fig 2c) has been well-studied (https://www.pnas.org/doi/full/10.1073/pnas.1610609114). Therefore, conservation at the level of these three major lineages lacks novelty (also has been published by the same authors by the references #9, #10). I also have other questions regarding the presented results (see below).
Major concerns:
1) The proposed network-dynamics analysis is based on ‘aggregated networks.’ I am not clear how to aggregate 15 clusters of human data and 36 clusters of mouse data (section 3.1) into 25 timepoints (page 9, line 304).
2) Additionally, the resultant (pseudo-)time points are dependent on not only the number of clusters identified but also the pseudo-time construction method applied. Given that bifurcating and trifurcating networks were hard to infer (Pratapa 2020 Mature Methods), I would like to see the robustness of the current conclusions under different parameters of cell clustering and trajectory construction (eg. slingshot predictions).
3) Gata 1 is known to be specifically activated during differentiation towards MEPs. In Figure 3a, the edge numbers along the MEP trajectory with 25 pseudotime points, present the lowest value at point 15 in human (top, middle) and at the pseudotime point 14 in mouse (bottom, middle). If the change of Gata1 expression is corresponding to the lowest edge number, I expect to see in Fig 3e, the expression changes happening in a window around 150 out of 250 pseudo bins. However, in Figure 3e, the human GATA1 expression changes in a window of 50th-80th bins, and the mouse Gata1 changes in a window of 100th-160th bins. In this regard, the proposed model (Fig 3b) is not supported finally. Therefore, more interpretation or explanation is required for Fig 3.
4) Page 13, line 428, the author state that ‘there was high correlation in genes’ connectivity values between human and mouse networks (Figure 4b).’ Statistically, a coefficient of 0.32-0.44 is at the edge to be significant. Can the author evaluate the significance for their argument?
5) Similarly, page 13, line 448, are the coefficients r=0.42, 0.24, and 0.26 significant?
6) Each subpanel of Figure 5 presents 5 columns x 3 rows comparisons between observation and random: network properties in columns and lineage-specific trajectory in rows. I notice that in most cases, the first 4 columns have a consistent pattern, but the last column presents a unique pattern. I wonder if the authors could interpret the consistency of their statistics.
Minimal concerns:
1) It is hard to follow when I come to Figure 1b with 3 parameters (d, h, A) without explanation in the legend or the corresponding text. In fact, these details are not required for Fig 1 to do a concept introduction.
2) The proposed method applies an existing method called loggle. I am wondering about the hypothesis and limitation of applying loggle, and then will scRNA-seq analyzed here (with multifurcations) meet these hypotheses?
3) In the introduction, page 3, line 83, I am lost when the authors mentioned ‘aggregated networks’ out of a sudden.
4) Figure fonts are too small (Fig 4, Fig 5) thus hard to see details.
5) Before introducing a new method, could the authors give the rationale? For example, page 13, section 3.7, why apply/choose DHI?
Reviewer 2 Report
Gao et al. analyzed pseudotime ordered HSC data to infer GRN evolution. It is an interesting question and the observations give some network-level insight (although not much mechanistic). Our key concerns and comments are as follows.
1. The authors chose some methods and ran a bioinformatic analyses. Recent benchmarking of scRNAseq pseudotime sorting methods have shown that monocle isn't always the best method. The authors should use at least one more method for pseudotime ordering to confirm the robustness of their conclusions.
2. For general datasets, pseudotime ordering can place very different cell types next to each other. Also, the branching structures for general datasets are not expected to be as clear as the HSC data. So, the approach described in this paper are not easily generalizable. The authors should clearly discuss this in the Introduction.
3. 1a: Is it data or a hypothetical example?
4. Equation 1, as presented, is not helpful. Either explain it or replace it with an intuitive description of what it is doing.
5. Line 160: what does neighborhood width mean? We strongly urge that loggle is explained in context of scRNAseq data. So, the notations should be easily relatable to scRNAseq data.
6. Explain what graph energies mean in context of GRNs
7. Line 303: it needs a clarification whether they obtained three loggle models for the three branches
8. Fig 4a: how were the clusters derived? Didn't this require any reordering of rows and columns?
Round 2
Reviewer 1 Report
The authors has addressed all my concerns.
I would suggest the following details to be added before publication:
1) Fig 1d, x-axis is confusing. Does 'pseudotie' mean 'the inferred pseudotime bins of cells' or 'pseudotime trajectory constructed of cell clusters'?
2) Fig 4b, please specify the statistic model for the p-value calculation (same for the other p-values in the paper).
3) line 204-395, can the authors provide p-values for the 20-30 times (8-15 times) higher?
Reviewer 2 Report
The authors have mostly addressed our concerns. Two remaining points are as follows.
Re: point 1 - When the authors say, "The relatively lower correlation in human may be due to the heterogeneity of different individuals and complicated lineage specification in human.", they need to clarify how many individuals they are talking about. Also the point "Complicated lineage specification" needs a citation.
Re: point 8 - explain the "manual" approach. This is not acceptable otherwise since the reproducibility is questionable.
